# Quantum Kernel Methods for Industrial Anomaly Detection

Takao Tomono [0000-0003-4984-4063]

Keio University, Yokohama, Kanagawa 212-0032 JAPAN
takao.tomono@ieee.org

**Abstract.** Anomaly detection plays a vital role in industrial quality control and manufacturing processes. Traditional machine learning methods often face challenges, especially in scenarios where training data is limited. In these circumstances, quantum machine learning (QML) has emerged as a promising approach to improve anomaly detection capabilities. This paper provides a comprehensive review of QML applications in industrial anomaly detection, with particular focus on image-based inspection systems, presenting our novel contributions. This paper classifies various types of anomalies encountered in industrial environments and provides a detailed review of classical and quantum anomaly detection approaches. In addition, we present the latest advances in quantum kernel methods in image-based anomaly detection. The analysis includes experimental results showing that quantum kernels outperform classical methods in certain industrial applications. For example, in shipment inspection, compared to an F1 score of 0.964 for SVM using an imbalanced dataset of 400 samples (300 normal, 100 anomalous), QSVM achieved an F1 score of 0.990 compared to 0.958 for ResNet (1132 normal), a 2.7% improvement in detection performance. We also discuss the implementation of quantum support vector machines (QSVM) with quantum kernels and their performance on quantum simulators and actual quantum hardware. Hardware validation reveals that quantum circuits with depths ≤32 maintain consistent performance between simulators and actual quantum devices, while circuits with depths >273 suffer significant degradation (AUC: 0.89→0.59) due to noise accumulation. These findings establish practical guidelines for deploying quantum machine learning in industrial settings and provide a roadmap for future quantum-enhanced manufacturing systems.

**Keywords:** Quantum machine learning, Anomaly detection, Quantum kernels, Support vector machines, Industrial inspection, Image processing.

## 1    Introduction

The manufacturing industry faces increasing demands for efficient and accurate quality control systems, particularly as product diversification leads to small-batch, high-mix production scenarios. Traditional machine learning approaches for anomaly detection often require large datasets for training, which can be challenging to obtain in specialized manufacturing contexts. Quantum machine learning (QML) has emerged as a

promising paradigm that leverages quantum mechanical principles to potentially overcome these limitations [1–3].

Quantum computing exploits phenomena such as superposition and entanglement to process information in fundamentally different ways than classical computers. It is believed that quantum computers can solve problems faster than classical computers. We adopt a different perspective on quantum computing advantages. In artificial intelligence and machine learning, quantum algorithms can explore exponentially large feature spaces more efficiently than their classical counterparts, potentially leading to improved classification performance with smaller training datasets. This capability is particularly relevant for industrial anomaly detection, where obtaining large numbers of defective samples for training can be impractical or costly.

The integration of quantum machine learning in industrial settings represents a significant step toward what has been termed the "sustainable smart factory with quantum technology." This concept encompasses not only improved efficiency in manufacturing processes but also reduced computational energy consumption through reversible quantum computation. As we progress through the NISQ era, practical applications of quantum machine learning are becoming increasingly feasible, with several successful demonstrations in industrial trial.[4–6]

This paper provides a comprehensive review of quantum machine learning applications for industrial anomaly detection [7–9]. We begin by examining the various types of anomalies encountered in industrial settings (Section 2), followed by a detailed review of both classical and quantum approaches to anomaly detection (Section 3). Section 4 presents our recent advances in image-based anomaly detection using quantum kernels, including experimental results from industrial applications. Finally, Section 5 discusses future perspectives and challenges in implementing quantum machine learning for industrial quality control. Within this review, we present the results of our work in Chapters 3.3 and 4.

## 2      Types of Anomaly Detection

### 2.1 Classification of Anomalies in Manufacturing

Anomaly detection in industrial contexts encompasses a wide range of defect types and detection scenarios. Understanding these different categories is essential for selecting appropriate detection methods and evaluating the potential benefits of quantum approaches. We examine point anomalies [10–12], contextual anomalies [13–18], and collective anomalies[19–22].

**Point Anomalies:**
These represent individual data instances that deviate significantly from the normal pattern. This anomaly is usually identified as an outlier, which is a significant deviation from the trend of the data. For example, if a certain credit card holder suddenly spends more on their credit card than usual, we investigate the possibility of credit card fraud.

In manufacturing, point anomalies are often the most straightforward to detect but can have significant implications:

Manufacturing Examples: A single defective product with a crack in an otherwise perfect batch, sudden spike in temperature readings during a stable process, individual measurement exceeding tolerance limits

Detection Characteristics: Point anomalies typically manifest as outliers in statistical distributions. They can be detected using threshold-based methods, statistical process control charts, or distance-based approaches

Challenges: Distinguishing between true anomalies and measurement noise, setting appropriate thresholds that balance sensitivity and specificity

**Contextual Anomalies:** Also known as conditional anomalies, these occur when data instances are anomalous in specific contexts but not others. This is when the normal behavior of a data point fluctuates depending on the situation. A common example is when a retail site experiences a large increase in traffic and sales on Black Friday, the busiest shopping day of the year. These spikes would be abnormal at other times of the year, so we set special parameters for this day.

Manufacturing Examples: High power consumption during machine startup (normal) vs. during idle state (anomalous), dimensional variations acceptable for prototype products but not for mass production, seasonal variations in environmental conditions affecting product quality.

Detection Requirements: Contextual anomaly detection requires understanding of operational states, temporal patterns, and environmental conditions. Methods must incorporate contextual features such as time of day, production phase, or product type.

Technical Approaches: Conditional probability models, time-series segmentation, multi-modal analysis combining sensor data with operational logs.

**Collective Anomalies:** These involve collections of related data instances that are anomalous when considered together. This is when a group of data points together exhibit abnormal behavior, even though each individual data point may appear normal. This anomaly can be identified by observing associations or patterns between multiple data points. A DDoS attack is an example of a collective anomaly, as it generates traffic from multiple sources that differ from the normal traffic pattern.

Manufacturing Examples: Gradual tool wear causing progressive quality degradation, coordinated drift in multiple process parameters indicating systemic issues, batch effects where entire production runs show subtle deviations.

Detection Complexity: Individual instances may appear normal, requiring analysis of patterns, trends, and correlations across multiple data points.

Detection Methods: Sequential pattern mining, change point detection, multivariate statistical process control.

### 2.2 Domain-Specific Anomaly Types

Industrial anomalies are classified into product anomalies (surface defects, dimensional variations, structural issues) and process anomalies (equipment degradation, parameter drift, environmental factors) [23–27]. This work addresses both standardized industrial

products [28–30] and non-standardized agricultural products[26,31,32], as demonstrated in our shipping inspection and apple quality control applications.
These are detailed as follows.

*Industrial Product:*
• Surface defects: scratches, dents, discoloration
• Dimensional anomalies: size variations, misalignment
• Structural defects: cracks, voids, inclusions
• Assembly errors: missing components, incorrect assembly
*Agricultural Product:*
• Internal defects: rot, hollow areas, internal browning
• External blemishes: bruises, insect damage, disease spots
• Shape irregularities: deformation, size variations
• Ripeness variations: over-ripe or under-ripe products
*Manufacturing (Growth) Process:*
• Equipment degradation: bearing wear, alignment issues
• Process parameter drift: temperature, pressure, speed variations
• Environmental anomalies: contamination, humidity effects

This detailed classification of content helps clarify the specific anomaly types and their detection requirements in each industry sector and allows the selection of appropriate detection methods.

## 2.3 Challenges in Industrial Anomaly Detection

Industrial anomaly detection presents several unique challenges that make quantum approaches particularly attractive as well as classical approaches. We describe the following challenges: Data Scarcity and Imbalance [33–36], High Dimensionality and Multimodality [37–40], Real-time Requirements [41–44], Dynamic and Evolving Patterns [45–49], and Cost Sensitivity [50,51].

**Data Scarcity and Imbalance:**
High-quality products dominate production, making defective samples rare and difficult to collect for training.

Statistical Challenge: In high-quality manufacturing, defect rates often below 0.1%, creating severe class imbalance
Economic Impact: Cost of producing defective samples for training purposes
Solution Approaches: Data augmentation, synthetic defect generation, transfer learning, few-shot learning techniques

**High Dimensionality and Multi-modality:**
Modern inspection systems generate high-resolution images and multi-sensor data. As a results, they generate multiple data types.

Visual: RGB images, hyperspectral data, thermal imaging
Physical: Force, torque, vibration measurements

Chemical: Spectroscopy, gas chromatography
Acoustic: Ultrasonic signals, acoustic emissions

**Real-time Requirements:**
Production line speeds demand rapid detection and classification.

Production Speeds: Modern production lines operate at speeds requiring sub-second decision making.
Latency Constraints: Detection, classification, and action must occur within tight time windows.
Edge Computing: Need for onsite processing capabilities rather than cloud based solutions.

**Dynamic and Evolving Patterns:**
Natural variations in materials and processes require robust detection methods.

Concept Drift: Normal patterns change over time due to tool wear, seasonal variations, or process improvements.
New Defect Types: Previously unseen anomaly patterns emerge with process or material changes.
Adaptive Requirements: Detection systems must continuously learn and update

**Cost Sensitivity:**
False positives and false negatives both carry significant economic impacts.

### 2.4 Anomaly Detection Metrics and Evaluation

When performing anomaly detection, it is sometimes difficult to know what evaluation metrics to use. Here, we will explain two important metrics.

Performance metrics that can be detected include true positive rate (sensitivity), false positive rate (1-specificity), precision, and F1 score. Classification Performance Metrics are formally defined as:

- True Positive Rate (TPR/Sensitivity/Recall): $TPR = TP/(TP+FN)$
- False Positive Rate (FPR): $FPR = FP/(FP+TN)$
- Precision: $Precision = TP/(TP+FP)$
- F1-score: $F1 = 2\times(Precision\times Recall)/(Precision+Recall)$

where TP, TN, FP, and FN represent true positive, true negative, false positive, and false negative classifications, respectively.

In addition, economic metrics include the cost of false positives (unnecessary exclusions) and the cost of false negatives (shipment of defective products). Meanwhile, operational metrics include the impact on throughput, inspection time, and computational resources.

As an evaluation challenge, there is the establishment of ground truthing techniques. This is due to the difficulty of obtaining accurate labels for all anomaly types. In

addition, as a time aspect, there is a trade-off between early detection and accurate classification. Finally, as a multi-class problem, different anomaly types require different actions.

## 2.5 Related work

**Classical Anomaly Detection in Industrial Setting:**
Traditional industrial anomaly detection has evolved from statistical control charts [52] to machine learning approaches, with SVMs becoming popular for small datasets. Recent surveys highlight the dominance of deep learning, particularly autoencoder-based methods for unsupervised anomaly detection[4,5]. However, these methods typically require substantial training data, limiting applicability in specialized manufacturing contexts with limited defective samples.

**Quantum Machine Learning for Pattern Recognition:**
Quantum machine learning addresses classical limitations through theoretical speedups. Rebentrost et al. [53] demonstrated quantum advantages for SVMs, while Havlíček et al. [2] provided experimental validation using quantum feature maps. Quantum kernel methods show particular promise for NISQ devices [54], with Huang et al.[55] proving that quantum kernels can achieve lower generalization error than optimal classical methods. However, most results assume ideal conditions, raising questions about practical applicability under noise.

**Quantum Anomaly Detection:**
Direct quantum applications to anomaly detection remain limited. Liu and Rebentrost [56] introduced quantum algorithms for anomaly detection with logarithmic scaling advantages. Corli et al. [1] surveyed quantum anomaly detection across supervised, unsupervised, and reinforcement learning paradigms. Industrial applications are largely unexplored, with Bhowmik and Thapliyal [57] discussing consumer electronics potential.

**NISQ-Era Limitation and Error Mitigation:**
NISQ devices face significant noise challenges [58]. Wang et al. [54] demonstrated that quantum advantages can vanish under realistic noise, with circuit depth as a critical factor. Some commercial solutions, including those involving error suppression, offer practical approaches to noise mitigation in near-term quantum devices.

**Positioning of This Work:**
This work addresses key gaps by tackling real-world industrial applications rather than synthetic datasets. Our systematic investigation of quantum kernel architectures (QK0-QK10) and comparative analysis of simulator versus hardware performance provides practical insights for NISQ-era deployment. The demonstrated quantum advantage in small-data regimes and applications across industrial and agricultural domains establish empirical evidence for quantum benefits in resource-constrained manufacturing environments. This work makes several distinctive contributions: (1) systematic investigation of quantum kernel architectures (QK0-QK10) with comparative performance

analysis, (2) empirical validation across both standardized industrial products and non-standardized agricultural products, and (3) comprehensive hardware validation demonstrating the practical constraints of NISQ-era quantum devices for industrial deployment.

## 3       Methods for Anomaly Detection

### 3.1 Classical Machine Learning Approach

Traditional machine learning methods have been extensively applied to anomaly detection in industrial settings. We provide a comprehensive analysis of each approach with mathematical foundations and practical considerations. We describe statistical method [8,52] and machine learning Algorithm [7,59,60] as following item.

**Statistical method:**
- Gaussian distribution modeling assumes normal data follows a Gaussian distribution, flagging instances beyond certain standard deviations as anomalies
- Regression analysis identifies anomalies as instances with high prediction errors
- Time series analysis methods like ARIMA detect anomalies in temporal data patterns

**Machine Learning Algorithms:**

*Support Vector Machines (SVM)*:
SVMs construct optimal hyperplanes to separate normal and anomalous classes. The kernel trick enables non-linear classification by mapping data to higher-dimensional spaces. Key advantages include effectiveness with small datasets and robust performance in high-dimensional spaces.

*Neural Networks*:
Deep learning approaches, particularly Convolutional Neural Networks (CNNs), have shown remarkable success in image-based anomaly detection. Autoencoders learn compressed representations of normal data, identifying anomalies through high reconstruction errors. However, these methods typically require large training datasets and significant computational resources

*Ensemble Methods*:
Random Forests and Gradient Boosting combine multiple weak learners to create robust anomaly detectors. Isolation Forests specifically designed for anomaly detection, isolate anomalies using fewer random partitions than normal instances.

*Clustering-Based Methods*:
K-means, DBSCAN, and other clustering algorithms identify anomalies as instances that don't belong to any cluster or form small, isolated clusters.

### 3.2 Quantum Machine Learning Approaches

Quantum machine learning leverages quantum mechanical principles to potentially achieve computational advantages over classical methods. Several quantum approaches have been developed for anomaly detection. Here, we describe an interface between PCA-quantum interface [61,62], QSVM [53,63,64], VQA [65–69], and quantum neural networks [70–72]to perform image quality control.

**PCA-Quantum Interface:**
The number of principal components directly determines the required number of qubits in quantum circuits. This creates a natural bridge between classical preprocessing and quantum processing, where PCA not only reduces computational complexity but also enables quantum encoding of classical industrial data.

**Quantum Support Vector Machines (QSVM):**
QSVMs utilize quantum kernels to compute inner products in exponentially large feature spaces. The quantum kernel function is expressed as:
$$\kappa(x_i, x_j) = \left|\langle\phi(x_i)^\dagger|\phi(x_j)\rangle\right|^2 = \left|\langle 0|U_k{}^\dagger(x_i)U_k(x_j)|0\rangle\right|^2$$
where $\phi(x)$ represents the quantum feature map, and $\kappa(x_i, x_j)$ is the quantum kernel function computing the inner product between quantum states $|\phi(x_i)\rangle$ and $|\phi(x_j)\rangle$. This approach has shown particular promise for problems with limited training data.
$\varphi(\mathrm{x})\colon \mathbb{R}^n \to \mathscr{H}$ represents the quantum feature map that encodes classical data into quantum Hilbert space $\mathscr{H}$, and the kernel computes the squared overlap between corresponding quantum states.

**Variational Quantum Algorithms (VQA):**
Variational Quantum Circuits (VQCs) combine quantum and classical processing, using parameterized quantum circuits optimized through classical optimization. The Quantum Approximate Optimization Algorithm (QAOA) can be adapted for anomaly detection tasks.

**Quantum Neural Networks:**
Quantum versions of neural networks leverage quantum gates as neurons and quantum states for information processing. These include Quantum Convolutional Neural Networks (QCNNs) for image processing and Quantum Autoencoders for unsupervised anomaly detection.

### 3.3 Comparative Analysis

There are two differences between classical and quantum machine learning. One is that a machine learning model can be built with a small data set [73–75], and the other is that the learning process itself is different and useful for anomaly detection [76].

The first point is that being able to build a machine learning model with a small data set is very useful for manufacturing small quantities of a wide variety of products in manufacturing sites.  Fig.1 illustrate performance of Quantum Kernel-SVM (Q-SVM)

compared to classical Kernel-SVM (C-SVM) and Resnet: a kind of Convolutional Neural Network (CNN)[75]. Quantum kernel is QK0 on Fig.4. The comparative performance of classical and quantum approaches is illustrated on across different metrics. Recent studies have demonstrated that quantum methods can achieve superior performance with smaller training datasets, making them particularly suitable for industrial applications where anomalous samples are rare. Methods for Anomaly Detection Natural variations in materials and processes require robust detection methods.

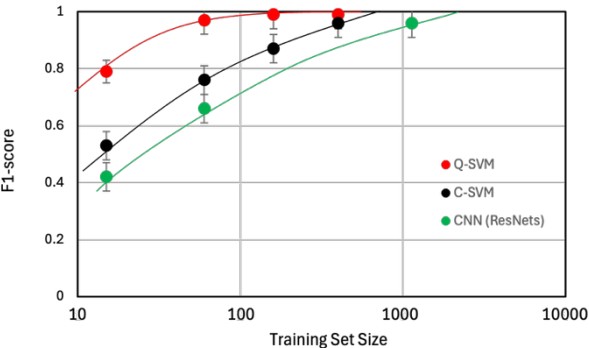

**Fig.1.** Quantum advantage in small-data regimes: F1-score performance comparison across varying training set sizes for industrial product classification. Experimental conditions: Real industrial product images, binary classification (normal vs anomaly), 15 principal components (The cumulative contribution rate is 0.82). Methods: Red circles: Quantum SVM (QK0 kernel), Black circles: Classical SVM (RBF kernel, $\gamma$=0.1), Green circles: CNN (ResNet-18). Key finding: Quantum SVM demonstrates superior performance for training sets <1000 samples, with F1-score advantage diminishing as dataset size increases. Statistical significance: Error bars represent means ± standard deviation across 3 independent runs. This figure shows the number of training sessions in logarithmic scale by adding some data to the numerical data in ref. [75].

Another point is that the learning process itself is different and effective for anomaly detection. Figure 2 shows the FPR and TPR values plotted in the ROC space using the Hearts disease and Iris_2 datasets. Here, we created a two-class dataset Iris_2 (versicolor and Virginia) with attribute 4 from the original Iris dataset. Figures (a) and (b) plot the training set size (ts) from 8 to 200 and 6 to 60 for hearts disease and Iris_2. The red and gray arrows show the hypotheses about the quantum and classical learning processes in the ROC space.

The gray line is how classical machine learning progresses and represents a random machine learning model. It is an ideal learning method that starts from TPR=0.5, FPR=0.5 and eventually progresses in the direction of TPR=1, FPR=1.

On the other hand, in the case of quantum machine learning, we observed that the learning process starts at a high TPR=1 and near FPR=1. If you maintain a high TPR, the FPR will decrease as the learning progresses (orange arrow in the figure). This means that the TPR is always 1, the inspection result is always a good product, and suspicious products are judged as defective. Therefore, no defective products are shipped to the market [76].

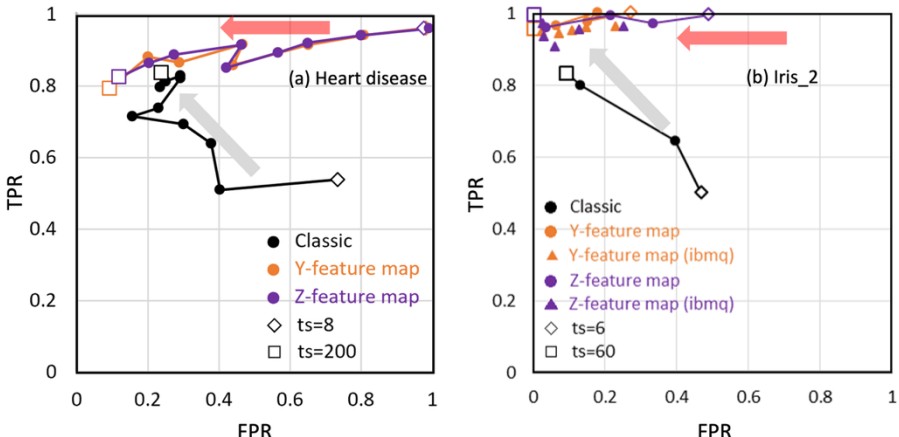

**Fig.2.** The plotting onto ROC space. Open diamond and square shape stand for training size of 8 and 200 on heart disease dataset, 6 and 60 on Iris_2 dataset. The red and gray directing arrow indicate hypothesis of learning progress. Ibm_Bogota is actual quantum computer [76]. TPR (True Positive Rate) and FPR (False Positive Rate) trajectories show different learning behaviors between classical and quantum approaches.

# 4      Image-based Anomaly Detection using Quantum Machine Learning

## 4.1 Quantum Kernel Design for Image Processing

The design of effective quantum kernels is crucial for image-based anomaly detection. Our research has explored various quantum circuit architectures, each offering different advantages.

Fig.3 shows the basic structure of a quantum kernel. It is represented by the inner product of $U_k(x_i)$ and its dagger. This calculated inner product is embedded as the kernel of the SVM. And then a machine learning model is constructed. Fig.4 shows the detailed circuit configuration of $U_k(x_i)$. We designed QK0 to QK10. The design concept is shown below [77].

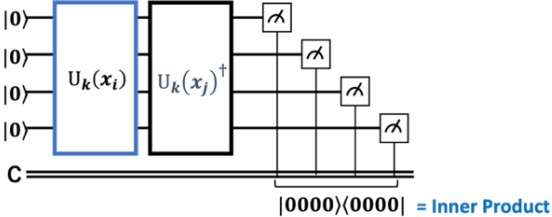

**Fig.3.** Quantum kernel circuits diagram in the 4qubits.

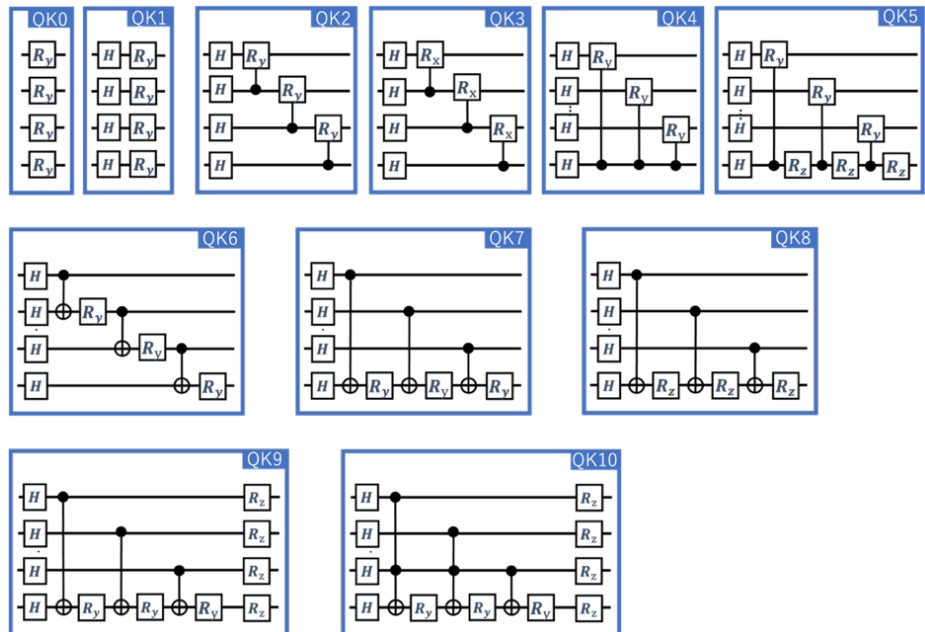

**Fig.4.** Quantum circuit architectures for different kernel designs. QK0 and QK1 are superposition only. QK2, QK3, QK4 and QK5 include control rotation gate. QK6, QK7, QK8, QK9, and QK10 include CNOTs and rotation gates.

Basic Rotation Gates (QK0, QK1): Simple circuits using Hadamard gates and single-qubit rotations provide baseline quantum kernels. While these show improvements over classical kernels, their expressivity is limited.

Controlled Rotation Gates (QK2-QK5): Circuits incorporating controlled rotation gates (C-$R_y$, C-$R_x$) create entanglement between qubits, enabling more complex feature relationships. These kernels show significant improvements in discriminative power.

CNOT-based Architectures (QK6-QK10): Advanced circuits using *CNOT* gates and combinations with rotation gates achieve the highest performance. The QK9 architecture, featuring *CNOT* gates connecting each qubit to a bottom qubit with Ry rotations between *CNOT*s, has proven particularly effective.

### 4.2 Industrial Application: Shipping Inspection

Machine learning is now widely used in shipping inspection at production sites. Products are quality controlled within a 3σ range to ensure that no basic defects occur. Conversely, as there are almost no defective products, machine learning models are constructed using unsupervised machine learning.

Like many factories, TOPPAN's factories also use unsupervised learning. To prevent defective products from being shipped, shipping inspection uses a three-stage model:

an image processing model, a discrimination model, and a generative model to prevent the outflow of anomaly products. This image-based shipping inspection process is shown in Fig.5. Products are shipped after passing through these three stages of inspection.

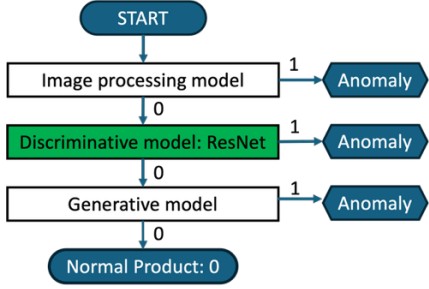

**Fig.5.** Overview of shipping inspection using classical machine learning at an actual production site. We use three model described in the figure actually [75]. Here, 0 and 1 stand for normal and anomaly.

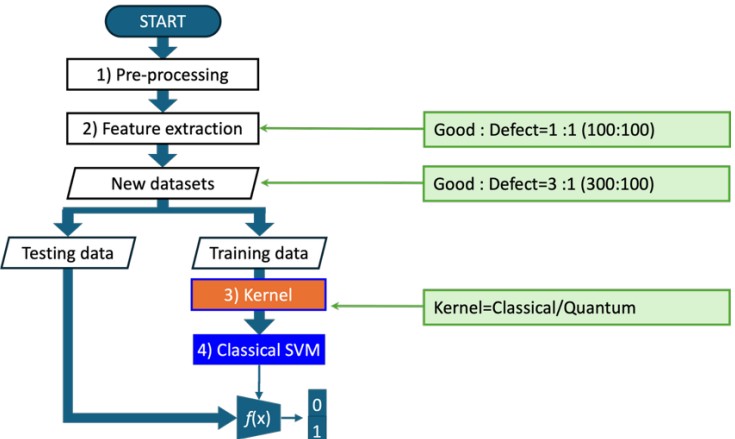

**Fig.6.** Application of supervised ML on shipping inspection in actual production sites. ML means machine learning [75].

**Table 1.** Performance in the shipping inspection.

| Model | Note | Datasets size | Accuracy | F1-score |
|-------|------|---------------|----------|----------|
| ResNets | Unsupervised learning | 1132 | 0.940±0.047 | 0.958±0.048 |
| C-SVM | supervised learning | 400 | 0.944±0.047 | 0.964±0.048 |
| Q-SVM | supervised learning | 400 | 0.988±0.049 | 0.990±0.050 |

Results represent mean ± standard deviation across 3 independent experiments. Original data is based on ref. [75].

Our implementation of quantum machine learning for shipping inspection demonstrates practical advantages in real-world settings. The system processes product images through several stages.

1. Image Preprocessing: Binarization and noise reduction.
2. Feature Extraction: Principal Component Analysis (PCA) to reduce dimensionality.
3. Quantum Kernel Generation: Creating quantum kernels based on selected features.
4. Classification: SVM with embedded quantum kernels.

Results show in the table 1. ResNet using unsupervised learning achieved an F1-score of 0.958 with nearly 1,000 training samples, while our quantum approach achieved superior results with an accuracy of 0.988 and of F1 Score of 0.990 with the quantum kernel using an imbalanced data set (300 normal, 100 anomalous). Even when using classical kernel RBF, both accuracy and F1 score were 0.944-0.958, which is not as good as the quantum kernel, but excellent results were obtained [75]. All experiments were conducted three times with different random seeds to ensure reproducibility, with variations remaining within 5% across all metrics.

### 4.3 Agricultural Product Inspection

Extending beyond industrial products, we applied quantum anomaly detection to agricultural products (apples with internal defects) [77]. Research into the use of quantum technology in the agricultural field [73,78] is also being conducted overseas, but there are challenges unique to the agricultural field. This application presents unique challenges:

- Non-standardized product shapes and sizes
- Internal defects invisible from external inspection
- Natural variations requiring robust classification

Fig. 7 shows the experimental setup for creating the dataset. Internal defects cannot be detected from external inspection. Therefore, we created a device that irradiates the apple with an LED from the top side and takes a photo from the bottom side. The resolution was 4032 x 3024. The photo taken from the top shifts to orange, but it is converted to a black and white image by binarization. Only the vine is captured, but if there is nectar, a spotted pattern appears. On the other hand, if there is a crack in the vine, it is not visible, but when enlarged, a black spreading area appears around the vine. This enlarged area was 120 x 80. In both cases, by cutting the apple in half and comparing the photo with the binarized image, it is possible to determine whether the apple is filled with nectar or has cracks.

Here, the apple we chose is Fuji. Fuji is the most widely produced apple variety in Japan and is characterized by a good balance of sweetness and sourness, and a good taste with nectar. For this reason, the appearance is also important, and internal defects lead to customer dissatisfaction, so it is best not to ship the product if possible. On the other hand, apples full of honey (which at first glance appear to be browning) are prized as they add to the flavor. However, cracks on the rind result in complaints.

Fig.8 shows the relationship between the F1 scores of the principal components of each quantum kernel used to identify internal vine cracks in apples, compared to classical kernel RBF. Fig.8(a) shows the results of the conventional kernel RBF and QK0-QK5. The horizontal axis is the feature corresponding to the cumulative contribution of the principal components. The vertical axis is the F1 score. When the feature is 3, it means the cumulative contribution of the first to third principal components. When the feature is 7, it means the cumulative contribution of the first to seventh principal components.

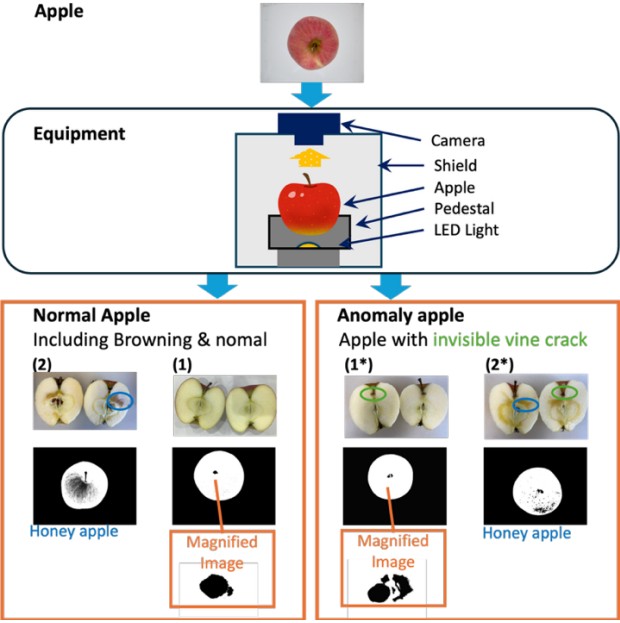

**Fig.7** Experimental setup for discriminating internal vine crack in the apple. Internal defects can be visualized using LEDs. Original image is 4032×3024. Magnified image is 120×80.

First, we compare the F1 scores of each kernel when the feature is 3, and then compare the trend of the F1 score as the features increases. The classical RBF increased by about 0.2 from the feature of 3 to 6, but there was almost no change when the feature size increased from 6 to 7. QK0 and QK1 are quantum circuits with one H gate, one H gate, and one rotation gate Ry for each qubit. When the features increased from 3 to 7, the F1 score value increased by more than 0.5. The F1 scores of QK2, QK3, QK4, and QK5 increase slightly as the feature value increases from 3 to 7, however, remain almost constant.

Fig.8(b) shows the results of QK6 to QK10 and the classical kernel RBF. When the features is 3, the F1 scores of QK7 and QK8 are more than 0.15 larger than RBF. QK6 shows almost the same trajectory as QK1 and QK2 (1), but QK7 and QK8 increase by 0.1 to 0.15 larger than RBF, even when the feature value increases, and by 0.15 larger.

Furthermore, the F1 scores of QK9 and QK10 are more than 0.3 larger than the classical kernel RBF when the feature value is 3, and the F1 scores also increase as the feature value increases.

From the above, it is thought that quantum circuits with entanglement can form more complex separation interfaces than superposition and rotation control gates. In particular, quantum circuits with an Rz rotation gate at the end are thought to be more effective. The F1 scores of QK9 and QK10 are the highest among these quantum kernels, and they are considered promising quantum kernel candidates.

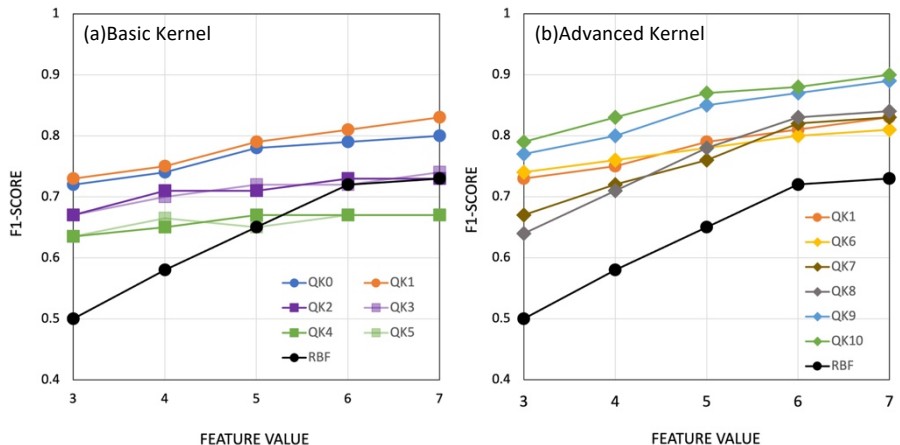

**Fig.8** Quantum kernel performance comparison for apple internal defect detection (vine crack). FEATURE VALUE means numbers of features (number of principal components). (a) Basic quantum kernels: QK0-QK5 vs classical RBF kernel. QK0/QK1 use simple rotation gates, QK2-QK5 incorporate controlled rotations. (b) Advanced quantum kernels: QK6-QK10 with CNOT-based architectures. QK9 and QK10 demonstrate substantial improvement over classical RBF, particularly with 3-7 principal components. Key observation: Quantum kernels with entanglement (QK6-QK10) significantly outperform both classical methods and simple quantum circuits.

## 4.4 Performance Analysis on Quantum Hardware

In the current analysis, it is better to evaluate the performance with a quantum simulator (ideal, noiseless simulation) and actual quantum hardware (IBM superconducting quantum processor with realistic noise). However, the availability of the quantum simulator depends on the environment, the user, and the conditions, and is subject to various constraints. In addition, the only way to use the quantum computer without error correction is to use it via the cloud. Basically, the data from the factory cannot be taken out of the factory. Therefore, the experiment with the quantum computer was only used for the shipping inspection of agricultural products.

Fig.9 shows the ROC curves comparing the quantum simulator and the quantum computer for the quantum kernels QK9 and QK10, which have 4 qubits and an AUC of 0.9 or more. Here, QC and QS mean the quantum computer and the simulator. The

left figure shows the ROC-AUC curves of RBF, QS, and QC for QK9, and the right figure shows the ROC-AUC curves of RBF, QS, and QC for QK10. For reference, we also plot the positions of the random model (black dashed line) and the ideal machine learning model (red dashed line). The dashed line from False Positive Rate (FPR) = 0, True Positive Rate (TPR) = 0 to FPR = 1, TPR = 1 shows the random model. The axis of FPR = 0 and TPR = 1 shows the ideal machine learning model. The AUC of the classical RBF is drawn close to the random model, and the numerical data is 0.62. In QK9, the behavior of the AUC curve on the quantum computer was the same as that of the quantum simulator. The numerical data of the AUC at that time was 0.90 for both, as shown in the figure. The difference in behavior between QK9 and QK10 is thought to be due to the difference in depth. The depths of QK9 are 32, while the depths of QK10 are 273, and it is thought that noise became unacceptable between these two.

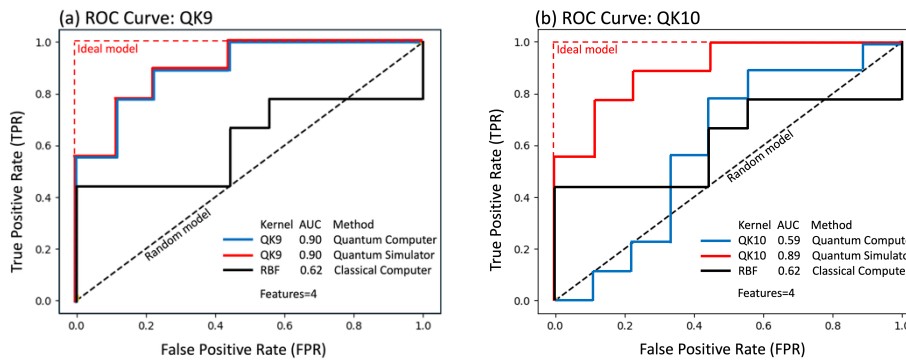

**Fig.9** ROC curve. Hardware validation: Quantum simulator vs actual quantum computer performance. Experimental setup: QK9 (left) and QK10 (right) kernels, 4 qubits, agricultural dataset. Hardware: IBM quantum computer vs ideal simulator. Performance metrics: QK9: AUC=0.90 (both simulator and hardware) QK10: AUC=0.89 (simulator) → 0.59 (hardware). Circuit depth impact: QK9 (depths=32) maintains performance, QK10 (depths=273) suffers noise degradation. Reference lines: Black dashed = random classifier, Red dashed = ideal classifier. Measurement: 1024 shots, 3 independent runs for hardware validation.

## 5      Future Perspectives

Fig.10 shows the industrial application roadmap of quantum technology in the manufacturing field. This figure is based on my experience in quantum computer-related research and development, including references [79–81] and other literature and books.

The quantum technology roadmap for industrial applications spans three phases: 1) Near-term (2025-2030), NISQ-optimized hybrid algorithms and specialized quantum kernels; 2) Medium-term (2030-2040) - on-premises quantum processors and real-time anomaly detection; 3) Long-term (2040+) - fault-tolerant quantum computers enabling fully quantum industrial control systems.

I am thinking of the overall picture of smart manufacturing using quantum technology in the future, as shown in Fig.11. System and product failure prediction, health

management, process optimization, energy management, production line, inspection system, and supply chain are centered on quantum computers and connected on-promise with classical computers.

I am thinking that integrating quantum processors with classical systems could improve processes by 10% to 20% and potentially make them 30%-50% more energy efficient.

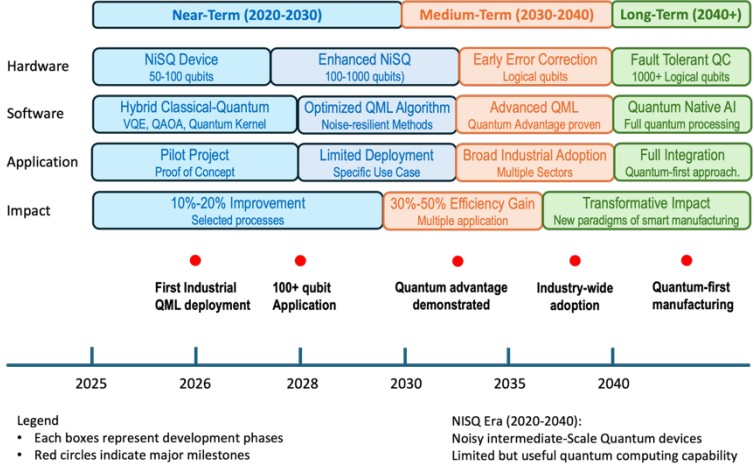

**Fig.10** Roadmap for smart manufacturing using quantum technology.

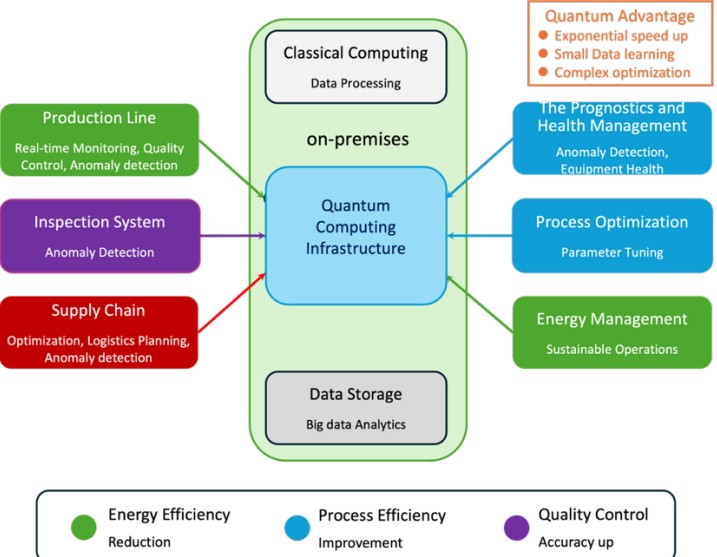

**Fig.11** Smart manufacturing using quantum technology in the future.

# 6     Conclusion

This comprehensive study has demonstrated the significant potential of quantum machine learning for industrial anomaly detection, establishing both theoretical foundations and practical applications for quantum-enhanced manufacturing quality control. Through systematic investigation of quantum kernel methods, we have shown that quantum approaches can achieve superior performance in small-data regimes typical of industrial settings.

Our research provides several critical contributions to the field. We have demonstrated quantum advantage in practical industrial scenarios, with quantum SVMs achieving F1-scores of 0.990 compared to 0.964 for classical SVM using 400 training samples. The systematic design and evaluation of quantum kernel architectures (QK0-QK10) revealed that CNOT-based circuits with controlled rotations significantly outperform simple rotation-based designs, providing concrete guidelines for quantum circuit design in industrial applications.

Empirical validation on actual quantum hardware provided crucial insights into NISQ-era limitations. Our analysis revealed that circuit depths $\leq 32$ maintain consistent performance between simulators and hardware, while depths $>273$ suffer significant degradation due to noise accumulation. This finding establishes practical constraints for current quantum implementations and highlights the importance of error mitigation strategies.

Deployment in real-world applications has achieved an F1 score of 0.990 (quantum simulator) for industrial product outgoing inspection and 0.90 (AUC: quantum computer) for agricultural product quality control. As shown above, we have demonstrated robust performance across a range of defect types, demonstrating the practical feasibility of quantum anomaly detection in manufacturing environments.

It is expected that this research will establish quantum machine learning as a transformative technology for industrial quality control.

Current NISQ devices show clear advantages in small-scale data processing but are limited in scalability and circuit complexity. In the future, on-premises infrastructure will be required, so efforts will be needed to break down the barriers to adoption in traditional manufacturing environments.

Future research should prioritize three key areas: (1) developing noise-resilient quantum algorithms for industrial deployment, (2) establishing standardized quantum-classical hybrid frameworks, and (3) creating scalable quantum infrastructure for manufacturing environments. The transition toward fault-tolerant quantum computing will unlock unprecedented capabilities for complex industrial optimization and quality control systems.

**Acknowledgments.** This work was supported by the New Energy and Industrial Technology Development Organization (NEDO) [Grant No. JPNP23003] and by the Center of Innovations for Sustainable Quantum AI (JST) [Grant No. JPMJPF2221].

**Disclosure of Interests.** The authors have no competing interests.

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
