# OpenReview forum: "Quantum Kernel Methods for Industrial Anomaly Detection"
_purdue.edu/Purdue_University/PQAI/2025/Symposium — PQAI 2025 Oral_

### Official Review · Reviewer_1duP · 2025-07-25
**A well-structured paper with practical relevance, but limited novelty and insufficient experimental transparency reduce its overall impact.**

**Rating:** 6
**Confidence:** 4

**Review:**

The paper presents a review and empirical evaluation of quantum kernel methods applied to industrial anomaly detection, targeting both manufacturing and agricultural quality inspection. The author surveys different types of anomalies, discusses classical and quantum approaches, and systematically evaluates various quantum kernel circuit architectures against classical machine learning models such as SVMs and ResNets. The work includes experiments on both simulators and real quantum hardware, aiming to demonstrate the practical advantages and limitations of quantum methods in real-world industrial applications where data is often scarce and high-dimensional.

Among its strengths, the paper clearly identifies the industrial challenges motivating quantum approaches and provides a broad and up-to-date review of relevant literature. The comparison of multiple quantum kernel architectures, along with the inclusion of results from actual quantum hardware, adds practical value. Additionally, the real-world case studies, especially those involving industrial and agricultural datasets, showcase the potential of quantum machine learning beyond purely theoretical or synthetic benchmarks.

However, the work has several notable weaknesses. First, the core empirical contributions are somewhat incremental, as much of the methodological content and performance claims build upon existing literature without significant novelty [it's not clear from the text that it's a review paper]. Second, experimental transparency is lacking, with insufficient detail on dataset accessibility, preprocessing, code, and hyperparameter choices, making reproducibility difficult. Furthermore, the discussion of results would benefit from a clearer theoretical foundation and a tighter focus on unique contributions rather than lengthy background sections.

---

### Decision · Program_Chairs · 2025-07-29

Accept (Oral)